# Effectiveness of Sinopharm’s BBIBP-CorV Booster Vaccination against COVID-19-Related Severe and Critical Cases and Deaths in Morocco during the Omicron Wave

**DOI:** 10.3390/vaccines12030244

**Published:** 2024-02-27

**Authors:** Jihane Belayachi, Abdelkader Mhayi, Hind Majidi, Elmostafa El Fahime, Redouane Abouqal

**Affiliations:** 1Acute Medical Unit, Ibn Sina University Hospital, Rabat 10000, Morocco; j.belayachi@um5r.ac.ma; 2Laboratory of Biostatistics, Clinical, and Epidemiological Research, Faculty of Medicine and Pharmacy, Mohammed V University, Rabat 10100, Morocco; 3Ministry of Health and Social Protection, Rabat 10100, Morocco; amhayi@sante.gov.ma (A.M.); hind.majidi222@gmail.com (H.M.); 4Molecular Biology and Functional Genomics Platform, National Center for Scientific and Technical Research (CNRST), Rabat 10102, Morocco; elfahime@cnrst.ma; 5Neuroscience and Neurogenetics Research Team, Faculty of Medicine and Pharmacy, Mohammed V University, Rabat 10100, Morocco

**Keywords:** booster dose, COVID-19, inactivated vaccine, SARS-CoV-2, third dose, vaccine effectiveness, omicron

## Abstract

Objective: This study investigates the effectiveness of the 1st booster dose against COVID-19 severe and critical hospitalizations and deaths occurring due to the Omicron wave in Morocco. Participants/methods: This study uses nationally representative data on COVID-19 from 15 December 2021 to 31 January 2022. The aim is to investigate the effectiveness of the inactivated COVID-19 vaccine BBIBP-CorV (Sinopharm) 1st booster dose against the Omicron wave in Morocco using real-world data established from nationally representative statistics on COVID-19 cases, deaths and vaccinations. Statistical Analyses: The screening method was used to estimate vaccine effectiveness against COVID-19 severe or critical hospitalization and COVID-19-related deaths. The data were grouped by, age subgroup, sex, week, and geographical area and were analyzed using binary logistic regression with an offset for vaccine coverage. Results: The overall BBIBP-CorV VE estimate is 89% (95% CI 85 to 92) effective in curbing COVID-19 deaths, and 81% (95% CI 78 to 84 in curbing COVID-19 severe/critical hospitalizations. Death-related VE estimate was 86% (95% CI 81 to 90) for patients aged ≥65 years, 96% (95% CI 90 to 98) for those aged <65 years, 95% (95% CI 88 to 98) in no-risk factor patients, 91% (95% CI 85 to 94) with 1 risk factor, 90% (95% CI 83 to 95) with 2 risk factors, and 72% (95% CI 52 to 84) in patients with 3 risk factors and more. Severe/critical hospitalization VE estimate was 78% (95% CI 74 to 82) for patients aged ≥65 years, 87% (95% CI 82 to 90) for those aged <65 years, 86% (95% CI 80 to 90) in no-risk factor patients, 80% (95% CI 73 to 84) with 1 risk factor, 80% (95% CI 70 to 85) with 2 risk factors, and 80% (95% CI 68 to 86) in patients with 3 risk factors and more. Conclusions: BBIBP-CorV boosters are effective in increasing protection against the Omicron variant-related COVID-19 deaths and severe/critical hospitalizations. The protection is reduced with older age and higher risk factors. These findings emphasize the importance of targeted vaccination strategies for different demographic groups and underscore the protective benefits of the first booster BBIBP-CorV vaccine.

## 1. Introduction

The coronavirus disease 2019 (COVID-19), caused by severe acute respiratory syndrome coronavirus 2 (SARS-CoV-2), has placed a significant burden on the healthcare system.

As of November 2023, the World Health Organization (WHO) has reported 771,679,618 confirmed cases of COVID-19, including 6,977,023 deaths, with a total of 13,534,457,273 vaccine doses administered [1]. In Morocco, as of November 2023, the MOH has reported the total number of contaminations to be 1,277,461 since the first case was reported there on 2 March 2020, with a weekly positivity rate of 6.3%. The total number of deaths is still 16,297 (general case fatality 1.3%). Challenges in public health persist due to the serial emergence of the SARS-CoV-2 variants [2].

In the current phase of the COVID-19 pandemic, it is crucial to conduct a comprehensive assessment of the effectiveness of vaccines against severe SARS-CoV-2 infections. In this context, it is essential to analyze the overall effectiveness of both the initial and the 1st booster dose, considering the specific vaccine types used. This evaluation contributes toward measuring the protection offered against various manifestations of COVID-19, with a particular emphasis on investigating the potential impacts of emerging concerning variants, including the Omicron variant (B.1.1.529). This analysis examines changes in vaccine effectiveness against these variants and their implications for public health strategies, with a specific concern for vulnerable populations such as the elderly, immunocompromised individuals, and those with chronic health conditions.

The primary variants, including the Omicron variant (B.1.1.529), exhibit distinct characteristics related to virulence, transmissibility, and vaccine evasion [3]. Research on immunogenicity suggests that vaccine effectiveness may decline over time; however, it remains unclear how this reduction translates to clinical vaccine effectiveness.

In a recent meta-analysis conducted by Wu et al., encompassing 68 studies on the long-term vaccine effectiveness of COVID-19 vaccines (mRNA and Non-Replicating Viral Vector) against infections, hospitalizations, and mortality, the authors noted significant decreases over time in vaccine effectiveness for SARS-CoV-2 infections, hospitalizations, and mortality for both the primary vaccine series and 1st booster dose. The Omicron variant exhibited lower levels of vaccine effectiveness at baseline, with further reductions over time [4]. Estimates of baseline vaccine effectiveness against hospitalizations in response to the Omicron variant were 71%, seven days after vaccination. The patterns of vaccine effectiveness change were similar for those vaccinated against the Omicron variant compared with the general data on any variant. The authors suggested that vaccine effectiveness findings align with data on immunogenicity, which indicates that the robust immunological response initially elicited by vaccination appears to diminish over time.

Regarding inactivated vaccines, a recent study by Huang et al. demonstrated that among vaccinated individuals, administering a third homologous dose, compared to receiving two doses of inactivated vaccine at least 181 days prior, was associated with a 60.5% reduction in the incidence of severe/critical illness and an 81.7% reduction in the incidence of COVID-19-related death. Booster vaccination provided the greatest protection associated with the intensity of control and prevention against COVID-19 epidemics [5].

Our previous study examined the primary regimen vaccine effectiveness of BBIBP-CorV (Sinopharm) against severe or critical hospitalizations due to COVID-19. This study found that the BBIBP-CorV vaccine is highly protective against serious SARS-CoV-2 infection in real-world conditions. Protection remains high and stable during the first three months following the primary dose but slightly decreases beyond the fourth month, especially in patients aged 60 years and older [6]. The rapid increase in COVID-19 cases resulting from the Omicron variant in vaccinated populations has raised concerns about the effectiveness of 1st booster dose of current vaccines against severe or critical cases requiring hospitalization, as well as against mortality.

Morocco has experienced three epidemic waves, each corresponding to distinct periods marked by the emergence of noteworthy SARS-CoV-2 variants. The first wave, occurring between February and May 2021, was characterized by the prevalence of the Alpha variant (B.1.1.7). The second wave, taking place from July to September 2021, witnessed the predominance of the Delta variant (B.1.617.2), accounting for 80% of the infections. The third wave, occurring during early January 2022, was characterized predominantly by a circulation of the Omicron (B.1.1.529) variant (Figure 1). The vaccination campaign was officially launched on 28 January 2021. Since 28 August 2021, BBIBP-CorV (Vero Cells) Sinopharm and BNT162b2 (Pfizer—BioNTech) vaccines have been deployed for the first booster doses.

Inactivated vaccines are under-studied in real-life conditions, despite being deployed in nearly 100 countries. The variations in vaccine effectiveness against the Omicron variant (B.1.1.529), especially for vulnerable populations, require further investigation.

Here, we present findings on vaccine effectiveness based on data obtained during the Omicron wave in Morocco. As of now, there is a scarcity of studies assessing the effectiveness of a BBIBP-CorV first booster dose against the Omicron variant in the adult population [5,7].

Hence, this study aims to assess the effectiveness of the inactivated COVID-19 vaccine BBIBP-CorV booster shots during the Omicron wave in Morocco. This evaluation relies on real-world data derived from nationally representative statistics on COVID-19 cases, fatalities, and vaccinations. Consequently, we seek to estimate vaccine effectiveness against severe or critical hospitalization due to COVID-19 and COVID-19-related deaths utilizing the screening method.

## 2. Methods

### 2.1. Study Design

Observational VE studies aim to emulate a randomized trial, in which vaccinated and unvaccinated individuals are comparable in their likelihood of being exposed to the virus and experiencing the outcome, apart from the key difference of whether they have received the vaccine. Observational studies cannot guarantee this comparability because vaccination is not randomly assigned, but they can attempt to approximate it using a variety of designs, such as cohort studies, case–control studies, test-negative case–control design (TND), and the screening method (case–population method) [8,9].

In this study, we estimate the real-world vaccine effectiveness (VE) against COVID-19-associated severe or critical hospitalized cases and COVID-19-related deaths between week 51/2021 and week 9/2022, after the administration of the booster doses of inactivated COVID-19 vaccine BBIBP-CorV using the screening method.

BBIBP-CorV is a whole inactivated (beta-propionolactone) adjuvanted (aluminum hydroxide) vaccine prepared from a Vero cells culture of the 19nCoV-CDC-Tan-HB02 (HB02) strain of SARS-CoV-2.

The screening method is a pseudo-ecologic design, which uses individual-level data on vaccination history from cases and vaccination coverage in the source population from which the cases came. The “screening method” approach uses estimates of [10] the following:(1)Cases: the proportion of persons with the disease who are vaccinated;(2)Reference population: the proportion of persons in the population who are vaccinated.

### 2.2. Data Collection: Cases

Clinical data concerning COVID-19 severe or critical hospitalization and death are registered in a dataset created for public health surveillance systems. Individual-level data collected on confirmed COVID-19 severe/critical hospitalizations and deaths during the period from 15 December 2021 to 31 January 2022 were identified.

For each case, data on age, sex, date of COVID-19 hospitalization, survival status, length of stay, specific clinical risk group status (the presence or absence of chronic respiratory disease, chronic heart disease, chronic kidney disease, diabetes, HTA or immunodepression) were extracted from the clinical surveillance systems.

For all sampled cases, vaccination status was extracted from the national register of vaccination (RNV) based on the national identity number. Vaccination status data included vaccine product received, number of doses, and administration dates.

The clinical information on COVID-19 was linked to the database of vaccinations, thereby providing both the date and the brand of the first, second, and the 1st booster dose of vaccination—or the lack thereof—for each infected person.

Patients were classified as follows:(1)Unvaccinated: not receiving any dose from any vaccine;(2)Primary vaccinated: 14 days after receiving either one of the doses of the recommended two-dose vaccine or a single dose of the Janssen vaccine, regardless of whether the person received a 1st booster dose;(3)1st booster dose: 14 days after receiving the 1st booster dose [11].

### 2.3. Data Collection: Reference Population

National vaccination register (NVR): Vaccination data were obtained from a common, unbiased electronic health record system. This dataset was based on a comprehensive and inclusive population-based list of target populations drawn up firstly for the entire population over the age of 17 based on a national identity card (NIC) (representing the general population from which the reported cases have emerged). This list is the basis of the NVR. Vaccine receipt is associated with the registration of vaccine information in the NVR.

The combined weekly and cumulative data on vaccine uptake, including the weekly number of first, second and 1st booster dose administered from each vaccine brand, stratified according to age groups (18–19, 20–24, 25–29, 30–34, 35–39, 40–44, 45–49, 50–54, 55–59, 60–64, 65–69, 70–74, 75+ years of age) were obtained from the RNV. To adjust for major confounders, the appropriate vaccine coverage was matched to cases at an individual level based on age group (18–19, 20–24, 25–29, 30–34, 35–39, 40–44, 45–49, 50–54, 55–59, 60–64, 65–69, 70–74, and 75+ years of age) and week of onset.

For a given week, data on population vaccination coverage two weeks prior to the week of case hospitalization or death was obtained including the period 15 December 2021–31 January 2022 (inclusive) corresponding to the Omicron wave. Population-level vaccination coverage was determined from this deidentified person-level COVID-19 vaccination data. Vaccination status was classified as described for hospitalized cases. Overall time between 1st booster dose receipt (28 august 2021) of the inactivated vaccine and hospitalization due to severe or critical manifestations of COVID-19 during the Omicron wave, as well as COVID-19-related deaths (1 January 2022) is within 15 to 120 days.

### 2.4. Inclusion and Exclusion Criteria

All patients that met the following criteria were included in the study: age ≥ 18 years; within the opportunity to receive the 1st booster dose of a COVID-19 vaccine; with hospitalization for severe or critical case; and RT-PCR laboratory-confirmed COVID-19 occurring within 14 days prior hospitalization.

Patients receiving only the first dose, a BNT162b2 (Pfizer-BioNTech) 1st booster dose, and those with any dose of COVID-19 vaccine received <14 days before hospitalization were excluded.

#### 2.4.1. Definitions

Severe and critical COVID-19: The WHO classifies severe COVID-19 as an individual infected with SARS-CoV-2 who exhibits an oxygen saturation of less than 90% while breathing room air and/or a respiratory rate exceeding 30 breaths per minute in adults, or displays signs of severe respiratory distress. Critical COVID-19 is defined as an individual infected with SARS-CoV-2 who develops acute respiratory distress syndrome, sepsis, septic shock, or other conditions that would typically necessitate the administration of life-sustaining therapies, including mechanical ventilation (invasive or non-invasive) or vasopressor therapy [12].

The opportunity to receive the 1st booster dose of a COVID-19 vaccine was assessed by the time between primary vaccination (two dose) and hospitalization. Severe or critical hospitalization case 5 months and more from the primary vaccination is considered within opportunity to receive the 1st booster dose. Only patients who received the BBIBP-CorV vaccine 1st booster dose were included regardless of the vaccine brand received for the primary vaccination.

#### 2.4.2. Ethical Considerations

This real-world study was pre-registered (https://osf.io/at3yf, accessed on 27 November 2023) on Open Science Framework. It was approved by the Rabat local ethics committee review board for biomedical research at Mohammed V University (N/21). A waiver of informed consent was granted for the study. Authorization N°A-RS-638/2021 from the National Commission for the Protection of Personal Data (CNDP) was obtained.

### 2.5. Statistical Analysis

Population vaccine effectiveness is defined as the reduction in disease risk among vaccinated versus unvaccinated persons in the population.

This screening method predicts the vaccine effectiveness against an outcome of a disease Y (COVID-19-related death and severe/critical COVID-19 hospitalization), for the vaccination status d (full or boosted vaccination, in all types of vaccine), using the following variables calculated collectively:

The proportion of the population vaccinated, PPV;

The proportion of outcomes within the vaccinated population, PCV.

Vaccine effectiveness against two outcomes of infections was determined using this method, namely against total severe/critical COVID-19 hospitalization and COVID-19-related death.

Vaccine effectiveness was calculated using the following formula:[1 − (PCV(1 − PPV)]/[(1 − PCV)PPV]

Screening method (SM) compares the vaccination coverage between reported cases and a reference group. This simple method was designed to be used as a rapid preliminary analysis when incidence and attack rate data are not yet available [13].

The data were grouped by age subgroup, sex, week, and geographical area and were analyzed using binary logistic regression with an offset (incorporating expected PPV by 1st booster dose and BBIBP-CorV for vaccine coverage). When estimating vaccine effectiveness for the 1st booster dose receipt, patients who had received one or two doses were excluded.

To obtain the adjusted VE estimates with 95% CIs, logistic regression was conducted with the outcome variable as the vaccination status of the case (PCV) and with an offset for the log-odds of the matched coverage. The same adjustment was included for population vaccination coverage.

We performed sensitivity analysis according to age group (65≥, <65 years) and risk factors count group (0: no-risk factor, 1: presence of 1 risk factor, 2: presence of 2 risk factors, and ≥3: presence of 3 risk factors or more) to estimate the COVID-19-related death and severe/critical COVID-19 hospitalization vaccine effectiveness.

*p* values were calculated to test the significance of the difference within each subgroup. A *p* value less than 0.05 was considered to be significant. For all point estimates of vaccine effectiveness, we calculated 95% CIs.

## 3. Results

Description of Cases

During the study period, 1741 patients were hospitalized for severe or critical COVID-19, and 699 cases were eligible for inclusion in the study as they met the eligibility criteria. The flowchart of this study is shown in Figure 2. Among the hospitalized patients, 35.3% received BBIBP-CorV (Sinopharm 1st booster dose; median age was 70 years, with 66.4% aged more than 65 years old. A risk factor was absent in 27.2% of the hospitalized patients. Mortality rate was 35.6%. Baseline characteristics of severe/critical COVID-19 hospitalization are summarized in Table 1.

## 4. Vaccine Effectiveness

### 4.1. COVID-19-Related Death VE

The overall VE estimate was 89% (95% CI 85 to 92). VE estimate was 86% (95% CI 81 to 90) for patients aged ≥65 years and 96% (95% CI 90 to 98) for those aged <65 years (Figure 3). According to risk factors count group, VE estimate in no-risk factor patients was 95% (95% CI 88 to 98), 91% (95% CI 85 to 94) with 1 risk factor, 90% (95% CI 83 to 95) with 2 risk factors, and 72% (95% CI 52 to 84) in patients with more than 3 risk factors (Figure 4).

### 4.2. COVID-19 Severe/Critical Hospitalization VE

The overall VE estimate was 81% (95% CI 78 to 84). According to age group, VE estimate was 78% (95% CI 74 to 82) for patients aged ≥65 years compared to 87% (95% CI 82 to 90) for those aged <65 years (Figure 3). According to risk factors count group, VE estimate in no-risk factor patients was 86% (95% CI 80 to 90), 80% (95% CI 73 to 84) with 1 risk factor, 80% (95% CI 70 to 85) with 2 risk factors, and 80% (95% CI 68 to 86) in patients with more than 3 risk factors (Figure 4).

### 4.3. Sensitivity Analysis

#### 4.3.1. COVID-19-Related Death VE

In patients aged <65 years old, the COVID-19-related death VE estimate was 98% (95% CI 93 to 99) in patients without any risk factors; 96% (95% CI 90 to 98) in patients with 1 risk factor, 96% (95% CI 89 to 98) in patients with 2 risk factors, and 88% (95% CI 70 to 96) in patients with 3 risk factors and more. In patients aged ≥65 years old, the COVID-19-related death VE estimate was 93% (95% CI 85 to 97) in patients without any risk factors, 89% (95% CI 81 to 93) in patients with 1 risk factor, 89% (95% CI 79 to 94) in patients with 2 risk factors, and 68% (95% CI 44 to 82) in patients with 3 risk factors and more (Figure 5).

#### 4.3.2. COVID-19 Severe/Critical Hospitalization VE

In patients aged <65 years old, the COVID-19 severe/critical hospitalization-related VE estimate was 90% (95% CI 84 to 93) in patients without any risk factors, 85% (95% CI 79 to 89) in patients with 1 risk factor, 85% (95% CI 77 to 98) in patients with 2 risk factors, and 85% (95% CI 76 to 91) in patients with 3 risk factors and more. In patient, aged ≥65 years old, the COVID-19 severe/critical hospitalization-related VE estimate was 84% (95% CI 77 to 89) in patients without any risk factors, 77% (95% CI 70 to 83) in patients with 1 risk factor, 76% (95% CI 66 to 83) in patients with 2 risk factors, and 77% (95% CI 65 to 85) in patients with 3 risk factors and more (Figure 6).

## 5. Discussion

We present the estimates of vaccine effectiveness (VE) for the third booster of the BBIBP-CorV vaccine within the framework of a nationwide mass vaccination campaign aimed at preventing severe and critical cases of COVID-19 hospitalization in the Kingdom of Morocco, with a specific focus on the Omicron VOC variant. Employing a screening methodology, we have ascertained that the administration of the BBIBP-CorV vaccine booster significantly reduces the risk of death (89%) and lowers the risk of severe/critical hospitalization (81%) due to COVID-19. This risk reduction is less pronounced in individuals with immune vulnerability associated with advanced age and comorbidities.

In individuals younger than 65 years old without any risk factors, there is a 90% reduction in the risk of severe or critical hospitalization. However, in older patients with one or more risk factors, this rate decreases to 77%. The risk of COVID-19-related death is reduced by 98% in patients aged under 65 years old without any risk factors. Nevertheless, in older patients with three or more risk factors, this rate of risk decreases to 68%.

While the effectiveness of COVID-19 vaccines has been extensively evaluated, there is limited data on how clinical risk factors impact vaccine effectiveness, especially in terms of its effectiveness against severe disease. Understanding the variations in vaccine effectiveness among these groups is crucial for guiding decision makers in formulating strategies for vaccine and antiviral prioritization.

In conclusion, our study demonstrates that individuals who received their 1st booster dose of the inactivated vaccine within 15 to 120 days before hospitalization due to severe or critical manifestations of COVID-19, as well as COVID-19-related deaths, exhibit vaccine effectiveness that aligns with the established guidelines provided by the World Health Organization (WHO). These guidelines specify that vaccine effectiveness against hospitalizations or mortality should meet a minimum threshold of 90%, accompanied by a lower 95% confidence interval of no less than 70%.

It is important to note that this level of vaccine effectiveness (VE) is not attained in patients aged over 65 who also have comorbidities. This emphasizes the necessity for targeted interventions and ongoing research to tackle the distinctive challenges encountered by this particular population group in achieving optimal vaccine protection. As we further enhance our comprehension of vaccine effectiveness across diverse demographics, these findings offer valuable insights for shaping public health strategies and vaccine distribution initiatives. In conjunction with the vaccination, it is imperative to consider policies related to interventions for vulnerable populations. These policies should encompass strengthening non-pharmaceutical interventions, such as mandatory face mask usage, the use of hand sanitizers and handwashing, as well as social distancing measures. Additionally, optimizing and prioritizing antiviral therapies should be emphasized.

Urquidi et al. demonstrated a positive correlation between full primary vaccination and consistent facemask usage, emphasizing the additional impact of facemasks in augmenting vaccine effectiveness to mitigate severe disease during periods of heightened viral circulation [14,15].

## 6. Future Directions and Conclusions

The timeliness of this research represents a significant strength, as it offers valuable insights into the most recent developments in the battle against COVID-19. This study tackles a highly pertinent and current issue: the efficacy of 1st booster dose of the BBIBP-CorV vaccine against the Omicron variant of SARS-CoV-2. By incorporating data from Morocco and focusing on the BBIBP-CorV vaccine’s effectiveness against the Omicron variant of SARS-CoV-2, this research provides a more comprehensive international perspective. The knowledge gleaned from this study may hold implications for other countries contending with similar challenges posed by the Omicron variant.

The effectiveness of BBIBP-CorV boosters in augmenting protection against Omicron variant-related COVID-19 deaths and severe/critical hospitalizations is evident. However, this protection diminishes with increasing age and higher risk factors. These findings underscore the significance of tailored vaccination strategies for diverse demographic groups and highlight the protective advantages of administering a third BBIBP-CorV vaccine booster.

It is crucial to note that vaccination remains an effective measure for reducing COVID-19 hospitalizations and fatalities. Nonetheless, other measures, such as face masks and physical distancing, may continue to be necessary for long-term infection control. Our findings offer valuable insights for clinicians, public health policymakers, and researchers concerning the long-term effectiveness of COVID-19 vaccines. These insights can inform clinical and policy recommendations, including considerations regarding the timing of future 1st booster dose.

## Figures and Tables

**Figure 1 vaccines-12-00244-f001:**
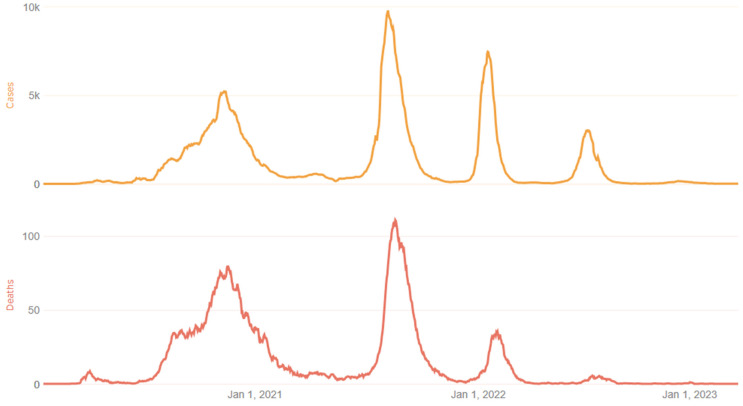
COVID-19 epidemic waves in Morocco [1]. National data of the three epidemic waves in Morocco. Each waves corresponding to distinct periods marked by the emergence of noteworthy SARS-CoV-2 variants. The first wave, occurring between February and May 2021, was characterized by the prevalence of the Alpha variant (B.1.1.7). The second wave, taking place from July to September 2021, witnessed the predominance of the Delta variant (B.1.617.2), accounting for 80% of infections. The third wave, occurring during early January 2022, was characterized predominantly by a circulation of the Omicron (B.1.1.529) variant.

**Figure 2 vaccines-12-00244-f002:**
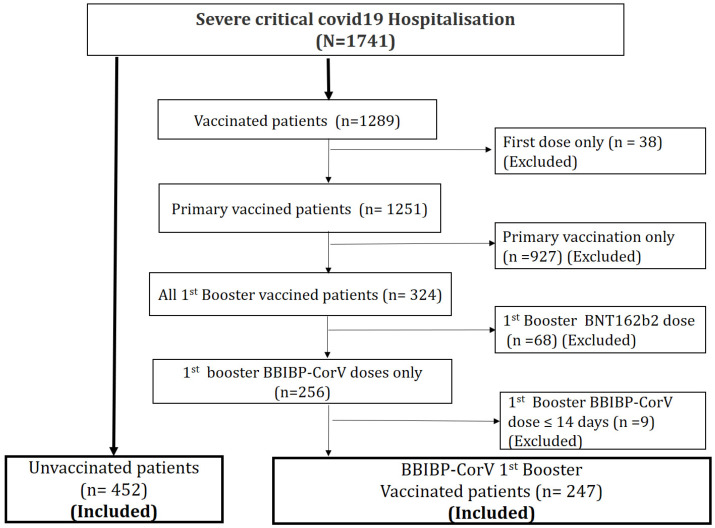
Flowchart of this study. A total of 1741 patients were hospitalized for a severe or critical form of COVID-19 during the Omicron wave. Among them, 699 were included in the study; 452 patients had never been vaccinated and 247 had received a 1st booster dose of the BBIBP-CorV vaccine.

**Figure 3 vaccines-12-00244-f003:**
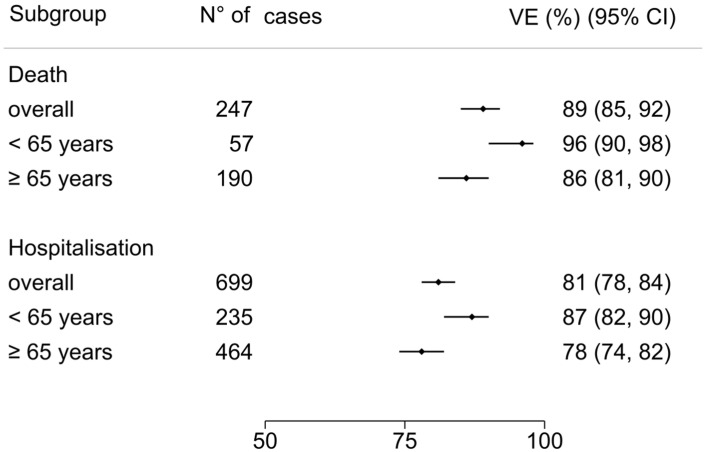
COVID-19-related death and hospitalization vaccine effectiveness for age subgroups: ≥65 years and <65 years. Error bars indicate the %[95% CI]. The overall VE COVID-19-related death and severe/critical hospitalization was 89% (95% CI 85 to 92) and 81% (95% CI 78 to 84), respectively. In age subgroups, death VE estimate was 86% (95% CI 81 to 90) for patients aged ≥65 years and 96% (95% CI 90 to 98) for those aged <65 years, and the severe/critical hospitalization VE estimate was 78% (95% CI 74 to 82) for those aged ≥65 years and 87% (95% CI 82 to 90) for those aged <65 years.

**Figure 4 vaccines-12-00244-f004:**
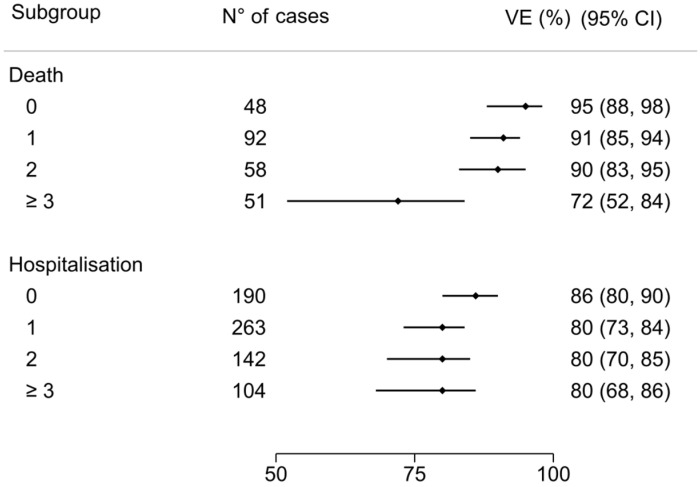
COVID-19-related death and hospitalization VE for risk factor count subgroups: death VE estimate in no-risk factor patients was 95% (95% CI 88 to 98); 91% (95% CI 85 to 94) with 1 risk factor; 90% (95% CI 83 to 95) with 2 risk factors; 72% (95% CI 52 to 84) in patients with more than 3 risk factors. The VE estimate among severe/critical hospitalization in no-risk factor patients was 86% (95% CI 80 to 90), 80% (95% CI 73 to 84) with 1 risk factor, 80% (95% CI 70 to 85) with 2 risk factors, and 80% (95% CI 68 to 86) in patients with more than 3 risk factors.

**Figure 5 vaccines-12-00244-f005:**
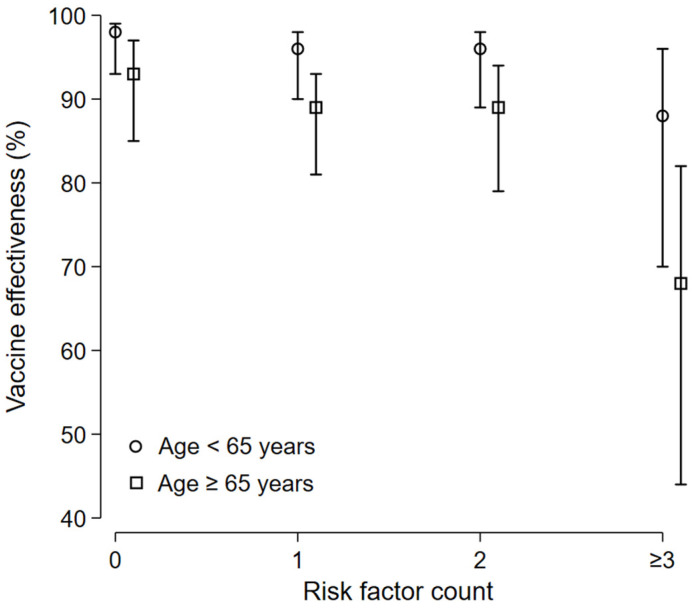
Death-related VE: subgroup analysis. In patients aged <65 years old, the COVID-19-related death VE estimate was 98% (95% CI 93 to 99) in patients without any risk factors, 96% (95% CI 90 to 98) in patients with 1 risk factor, 96% (95% CI 89 to 98) in patients with 2 risk factors, and 88% (95% CI 70 to 96) in patients with 3 risk factors and more. In patients aged ≥65 years old, the COVID-19-related death VE estimate was 93% (95% CI 85 to 97) in patients without any risk factors, 89% (95% CI 81 to 93) in patients with 1 risk factor, 89% (95% CI 79 to 94) in patients with 2 risk factors, and 68% (95% CI 44 to 82) in patients with 3 risk factors and more.

**Figure 6 vaccines-12-00244-f006:**
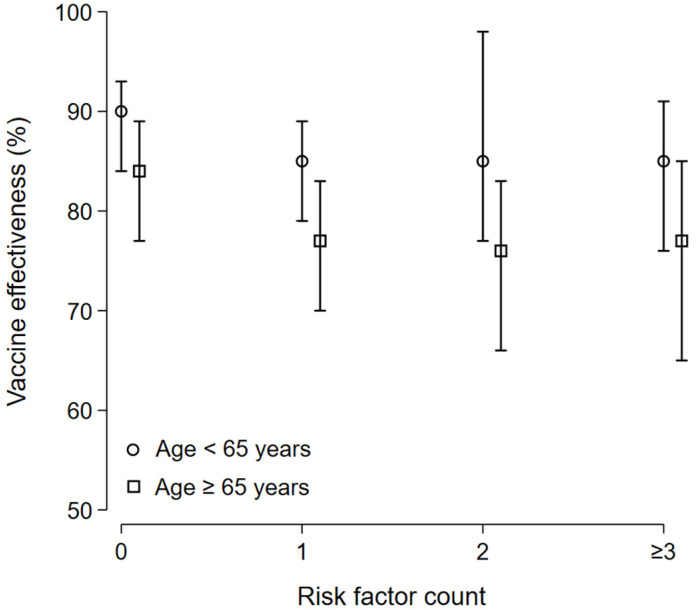
COVID-19 severe/critical hospitalization-related VE: subgroup analysis. In patients aged <65 years old, the COVID-19 severe/critical hospitalization-related VE estimate was 90% (95% CI 84 to 93) in patients without any risk factors, 85% (95% CI 79 to 89) in patients with 1 risk factor, 85% (95% CI 77 to 98) in patients with 2 risk factors, and 85% (95% CI 76 to 91) in patients with 3 risk factors and more. In patients aged ≥65 years old, the COVID-19 severe/critical hospitalization-related VE estimate was 84% (95% CI 77 to 89) in patients without any risk factors, 77% (95% CI 70 to 83) in patients with 1 risk factor, 76% (95% CI 66 to 83) in patients with 2 risk factors, and 77% (95% CI 65 to 85) in patients with 3 risk factors and more.

**Table 1 vaccines-12-00244-t001:** Baseline characteristics of severe/critical COVID-19 hospitalization.

	Overall Severe/Critical COVID-19 Hospitalization during Study Period (N = 1741)	Overall Severe/Critical COVID-19 Hospitalization Included in the Study (N = 699)
Sex		
Female	774 (44.5%)	
Male	967 (55.5%)	
Age median [IQR]	69 [60–79]	70 [61–80]
Age subgroup		
≤39 years	149 (8.6%)	64 (9.2%)
40–59 years	278 (16.0%)	89 (12.7%)
60–64 years	201 (11.5%)	82 (11.7%)
≥65 years	1113 (63.9%)	464 (66.4%)
Risk factor count		
0	481 (27.6%)	190 (27.2%)
1	673 (38.7%)	263 (37.6%)
2	347 (19.9%)	142 (20.3%)
3	240 (13.8%)	104 (14.9%)
Comorbidities		
0	481 (27.6%)	190 (27.2%)
1	1260 (72.4%)	509 (72.8%)
Risk factor		
Diabetes	553 (31.8%)	227 (32.5%)
Hypertension	598 (34.3%)	271 (38.8%)
Chronic obstructive pulmonary disease and Other respiratory disease	289 (16.6%)	100 (14.3%)
Cancer	238 (13.7%)	100 (14.3%)
Heart disease	327 (18.8%)	147 (21.0%)
Chronic kidney disease	167 (9.6%)	60 (8.6%)
vaccine statut		
Unvaccinated	452 (26.0%)	452 (64.7%)
Only first dose	38 (2.2%)	
Primary vaccinated	927 (53.2%)	
First booster dose	324 (18.6%)	247 (35.3%)
Survival status		
0	1178 (67.7%)	452 (64.4%)
1	563 (32.3%)	247 (35.6%)
Length of stay (by days) Mean (±SD)	6.5 ± 4.6	6.4 ± 4.7
Primary vaccination product		
BBIBP-CorV	457 (36.5%)	----------
BNT162b2	5 (0.4%)	----------
ChAdOx-1 S	789 (63.1%)	----------
First booster dose vaccine product		
BBIBP-CorV	256 (79.0%)	
BNT162b2	68 (21.0%)	----------

## Data Availability

The datasets generated for this study are available on request from the corresponding author.

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
