# Peer review of "Effectiveness of Sinopharm’s BBIBP-CorV Booster Vaccination against COVID-19-Related Severe and Critical Cases and Deaths in Morocco during the Omicron Wave"

_vaccines, 2024, doi:10.3390/vaccines12030244_

Round 1
Reviewer 1 Report
Comments and Suggestions for Authors
21: Morocco capitalize
49: the citation style is pretty different from every MDPI article I’ve read. Typically they follow Chicago style
87: check tense, especially for previous studies, they should be past tense
93 and Figure 1 (you have it labelled as figure 2): Need references for this data. Figure legend is not adequate, it should function on its own
101: capitalize
104: spacing
Methods: this entire section needs to be reworded for clarity. Also a lot of grammar and capitalization issues, and 164 starts in the middle of a sentence. It is very hard to follow your study design. It is unclear where this data was actually collected, I understand there’s some kindof national database. So you took patients who had severe disease, then cross-referenced that with a national database of vaccine status. How were p values calculated? What statistical tests are you using, and what software?
Figure 2 (labelled figure 1): legend needs more information
Figure 5 and 6: your graph and in-text data don’t match up (your 3-risk factor patient % don’t match up with the text)
332: French
Conclusions: it’s difficult to take your conclusions at face value when figures and in-text data are not lining up, I don’t trust the data set
Comments on the Quality of English LanguageNeeds some grammar and spelling updates
Author Response
Thank you very much for taking the time to review this manuscript. Please find the detailed responses below and the corresponding revisions/corrections highlighted/in track changes in the re-submitted files

Reviewer 2 Report
Comments and Suggestions for Authors
This manuscript delineates the Effectiveness of the inactivated COVID-19 booster vaccination that could reduce the life-threatening conditions. The data on inactivated COVID-19 in the literature is scant, and this manuscript could be a piece of this information.
As with other vaccine types, multiple doses and/or recent could protect more than less dose and/or received for a long time (waning immunity). The outcomes in this manuscript are consistent with other publications. The analyses in this manuscript are suitable. The quality of this manuscript is acceptable after revision.
Please check the format of citations in your manuscript. The numbering citation is required for this journal, not the Roman numbering.
Major concerns.
1. Please add the reference(s) that you used data to visualise in Figure 2.
There were national data or simulations?
Please clarify it.
2. Line 142. What about the off-labelled interval between 1st and 2nd dose?
Such as;
1) the interval between 1st and 2nd dose less than 3-4 weeks that decreases the immunity, or
2) the interval between 1st and 2nd dose of more than 3-4 weeks that increases the immunity due to the anamnestic response.
Did you include weird intervals in the analyses?
3. Did you include any immunocompromised in the analyses?
Minor concerns.
1. Table 1 "Third booster dose". This term may lead to confusion because the third booster means full vaccination (1 or 2 doses depending on the vaccine) with 3 booster shots
It means 4 or 5 doses.
Suggest using 1st booster or 3 doses to make it clear.
2. Have you tried to subgroup analysed by the interval between the last dose and outcomes (hospitalisation or death)? Such as recently (14 days - 1 month), 2-3 months and> 3 months.
Late vaccination may reduce the VE due to the immunity waning.
Comments.
1. Lines 46-48. Suggests adding a certain date of total cases and deaths that you were cited.
2. This manuscript mentioned the BNT162b2 with the manufacturer's name in the parenthesis (Pfizer—BioNTech). However, the BBIBP-CorV mentioned a cell line for culture in the parenthesis (Vero cells).
There were inconsistent formats.
Suggest using BNT162b2 (Pfizer—BioNTech) and BBIBP-CorV (Sinopharm) to make it consistent.
You may mention the cell line (Vero cells) in the study design with a bit of vaccine information (e.g. cell lines, virus strain, posology, or inactivating agent).
3. Suggests using "COVID-19" instead of covid 19, covid-19 or any variation. Because this word is an abbreviation, all capital letters are required.
For example, lines 121, 132, 148, 150, 159, 166 or elsewhere.
4. Line 150. Suggest using "RT-PCR" instead of rt-Pcr.
5. Line 164. Suggest using BNT162b2 instead of Pfizer.
6. Figure 1 and Table 1. Suggest using BNT162b2, BBIBP-CorV or ChAdOx-1 S instead of Pfizer—BioNTech, Sinopharm or Oxford—AstraZeneca.
7. Line 332. What is the French term in this line?
Author Response

(The authors gave the same response as above.)

Round 2
Reviewer 1 Report
Comments and Suggestions for Authors
You made a lot of great changes so thanks for that. You still have capitalization issues (morocco again). Do a word search to find them.
Also, you spelled introduction wrong.
You improved your reference style but some are still after the period, they need to go before. You need to go over each ref in the paper and make sure its right
Comments on the Quality of English LanguageMentioned above, some minor issues but they need to be dealt with
Author Response
Dear Reviewer 1,
Thank you for your constructive feedback. I appreciate your acknowledgment of the changes made. I thoroughly address the capitalization issues by conducting a word search to identify and rectify them.
I apologize for the misspelling of "introduction" and will make the necessary correction.
Regarding the reference style, I will ensure that each reference is correctly placed before the period. I understand the importance of maintaining consistency in citation formatting, and I will carefully review each reference in the paper to ensure accuracy.
Thank you for pointing out these areas for improvement. I will diligently address these issues to enhance the overall quality of the paper.
Best regards,
Reviewer 2 Report
Comments and Suggestions for Authors
This version is acceptable. The effectiveness reflects the actual situation of the vaccine used in the real world but has a lot of limitations in the outcomes.
1. Suggest using COVID-19 instead of "Covid-19", "COVID 19", or any variations throughout the manuscript because this term is an abbreviation. All letters must be in capital letters.
I found many lines used "Covid-19", including line 357. Please check it.
2. You already mentioned the manufacturer name of the BBIBP-CorV in lines 91-92, "BBIBP-CorV (Sinopharm)".
After this line, you can use only the vaccine name "BBIBP-CorV" without mentioning the manufacturer's name.
Author Response
Dear Reviewer,
Thank you for your insightful comments on our manuscript. We appreciate your careful review and have addressed the issues you raised. Below are our responses to each of your comments:
We acknowledge your suggestion to use "COVID-19" consistently as it is an abbreviation, and we have carefully revised the manuscript accordingly. All instances of "Covid-19" have been updated to "COVID-19," including the occurrence on line 357.
Thank you for pointing out the redundancy in mentioning the manufacturer's name after introducing "BBIBP-CorV (Sinopharm)" in lines 91-92. We have revised the manuscript as per your recommendation, and moving forward, we use only the vaccine name "BBIBP-CorV" without mentioning the manufacturer's name.
We hope these revisions address your concerns adequately. If you have any further suggestions or comments, please feel free to let us know. Your feedback is invaluable in improving the quality of our work.
Thank you once again for your time and thoughtful review.
Best regards,